# Study protocol for a mixed methods convergent investigation of family domestic and sexual violence in multiple sclerosis and broader neurology in Australia

Cassie Nesbitt[1,2,3]*, Genevieve Rayner[1,2,4], Kristen Timmens[5], Christina Kazzi[1,2], Rubina Alpitsis[1,2], Emma Foster[1,2], Jason Ray[1,2], Joanne Crosby[2], Helmut Butzkueven[1,2], Anneke Van Der Walt[1,2], Vilija G. Jokubaitis[1,2]

**1** Department of Neuroscience, Monash University, School of Translational Medicine, Melbourne, Victoria Australia, **2** Department of Neurology, Alfred Health, Melbourne, Victoria, Australia, **3** Department of Neurology and Neurosciences, University Hospital of Geelong, Geelong, Victoria, Australia, **4** Melbourne School of Psychological Sciences, The University of Melbourne, Parkville, Australia, **5** Consumer Advisor, Alfred Health, Melbourne, Victoria, Australia

* cassie.nesbitt1@monash.edu

## Abstract

### Background

People with disabilities experience disproportionately high rates of family, domestic, and sexual violence (FDSV), yet recognition and response in neurological healthcare, including multiple sclerosis (MS), remain underexplored. The study is informed by evidence-based frameworks for healthcare responses to family, domestic, and sexual violence, including the World Health Organization's *Listen, Inquire, Validate, Enhance safety, Support access* (LIVES) approach and Victoria, Australia's Multi-Agency Risk Assessment and Management (MARAM) framework. These models share core principles of trauma-informed practice, structured risk assessment, and coordinated response, providing a foundation for investigating how such approaches can be applied and adapted within neurological care.

### Objectives

This study aims to estimate the prevalence and describe the forms of FDSV experienced by people with MS, identify associated social, clinical, and contextual factors, and document current neurology practices for recognising and responding to FDSV. Comparative qualitative analysis across MS, epilepsy, and headache will assess whether findings are condition-specific or shared across neurology to inform future strategies.

**Data availability statement:** Individual level data from this study cannot be shared publicly because it concerns family, domestic, and sexual violence and may contain information that could enable re identification and create safety risk. Risk is dynamic and cannot be assumed to remain low over time, and the single centre design and low cell sizes increase re identification risk. These restrictions are imposed under the conditions of ethics approval by the Alfred Health Human Research Ethics Committee and the Monash University Human Research Ethics Committee. Any release of de identified, aggregated outputs is subject to approval by both approving ethics committees. Data requests can be submitted to the Alfred Health MS and Neuroimmunology Research team responsible for data management at msniresearch@alfred.org.au.

**Funding:** The author(s) received no specific funding for this work.

**Competing interests:** The authors have declared that no competing interests exist.

## Methods and analysis

A convergent mixed-methods design will be applied across six study arms. Quantitative components include screening approximately 1,000 people with MS, registry linkage, and surveys with at least 100 participants assessing trauma exposure, wellbeing, and disability-related violence. Qualitative interviews with people with MS, epilepsy, and headache, and clinicians (n ≈ 64 total) will explore experiences, clinical practice, and barriers and enablers to implementing evidence-based frameworks. Data will be integrated through triangulation to identify convergent and divergent patterns.

## Discussion

Findings will provide foundational evidence on how MS and other neurological conditions intersect with experiences of FDSV, guiding trauma-informed training, policy, and practice across neurology.

## Registration

Australian New Zealand Clinical Trials Registry (ANZCTR), ACTRN390279.

---

## Background

Violence is a widespread and pervasive problem. In Australia, an estimated 41% of adults aged 15 and over have experienced physical or sexual violence, reflecting similar prevalence internationally [1]. Women with disabilities experience violence at about twice the rate of non-disabled women, often with greater severity, and risk increases with the number and level of disabilities [1,2]. Men and gender-diverse people with disabilities are also disproportionately affected, though data is more limited [1].

Although intimate partner violence (IPV) is the most reported form among women with disability, a sole focus on IPV overlooks the overlapping and broader spectrum of perpetrators and forms of violence experienced by people with disabilities and across genders [1]. In Victoria, family, domestic, and sexual violence (FDSV) encompasses intimate, familial, kinship, and care-based violence, a framing adopted in this study [3]. (Box 1) Violence in disability contexts can involve both general and disability-specific forms, including neglect, withdrawal of care, exploitation, psychological violence targeting disability, restriction of financial autonomy, and exclusion from decision making [2].

Structural and social factors such as ableism, discrimination, isolation, and reliance on others for care contribute to environments where people with disabilities are targeted and face barriers to autonomy and support [2,4]. This is shaped by intersectionality across disability, gender, socioeconomic position, and other forms of disadvantage [5]. Language that labels people with disabilities as "vulnerable" reinforces stereotypes and shifts focus away from perpetrators [6]. A more accurate framing

recognises that risk is created through systemic conditions and intersecting disadvantage, rather than being inherent to disability [2,4].

## Research context

Multiple Sclerosis (MS) is characterised by demographic and clinical features that align with recognised risk factors for violence, including onset during reproductive years, predominance among women, socioeconomic disadvantage, comorbid mental health conditions, and progressive disability across the disease course [5,7]. Despite this alignment, MS-specific evidence remains insufficient to determine whether people with MS experience higher rates of violence [5,8].

The MS focused literature is international (United States, [9–12] Iran, [13,14] Italy, [15] Sweden, [16] and Norway [17]) but remains small and methodologically heterogeneous, with substantial variation in definitions, perpetrator scope, measurement tools and study design. Further, most studies rely on self-reported cross-sectional data with limited adjustment for confounders [5,8]. The only population-based comparator study, from Norway, included 106 pregnant women with MS and 77,278 controls, and reported higher adjusted odds of emotional violence (aOR 1.61, 95% CI 1.03 to 2.53), systematic humiliation (aOR 1.75, 95% CI 1.08 to 2.83), rape (aOR 2.37, 95% CI 1.02 to 5.49), and revictimisation (aOR 2.23, 95% CI 1.22 to 4.10) in women with MS [17]. Although restricted to pregnancy and limited by small case numbers, it provides initial evidence that some forms of violence may be more common in MS.

Prevalence estimates vary widely because studies assess different constructs within different sampling frames and perpetrator scopes, which limits direct comparability across cohorts [5,8]. In SocialMS (66 Italian centres, n = 1,004), 23.4% of people with MS reported any lifetime abuse, most commonly emotional or psychological abuse (20.2%), followed by physical (5.6%), financial (4.2%), and sexual abuse (2.5%) [15]. SocialMS also captured perpetrator context. Among respondents reporting emotional or psychological abuse who provided perpetrator relationship details, workplace contexts (39.6%), partner relationships (32.1%), and other family members (27.4%) were reported, and categories were not mutually exclusive [15]. SocialMS further reported clustering of abuse reporting with clinically relevant burden, including higher smoking, comorbidities, financial strain, and moderate or greater disability, with disability remaining associated in multivariable modelling using patient determined disease steps (OR 1.18), alongside progressive phenotype, longer disease duration, and more treatments in univariable models [15].

Other MS studies have used narrower perpetrator scopes or sampling frames. A North American registry survey focused on mistreatment by unpaid caregivers, where 54.9% reported any mistreatment and subtypes included psychological abuse (44.2%), financial abuse (25.2%), neglect (16.5%), physical abuse (11.2%) and sexual abuse (8.3%) [12]. A separate North American study combining prospective screening during routine neurology visits with retrospective mental health note review reported physical and or sexual abuse in 20% to 36% across two phases, repeated episodes in 84% of disclosures, and 38% of women reporting violence despite no cases being documented in clinical records, highlighting failures to recognise and respond within clinical care [9]. Two Iranian single centre cross-sectional studies in married women with MS reported partner violence in roughly half of participants, with one finding any IPV in 51.8% (physical 31%, emotional 27.5%, sexual about 9%) and the other reporting domestic violence domain estimates of 63% economic, 53% psychological, 34% physical and 20% sexual [13,14].

Evidence linking violence exposure to MS outcomes remains limited. Most MS studies do not establish when violence exposure occurred relative to MS onset, diagnosis, or disability accumulation, which limits inference about directionality and constrains interpretation of any associations with relapse activity or disability severity. In a small Swedish cohort (n = 47, relapse onset MS), adult violence exposure was associated with higher CSF IL 6, but the association attenuated after accounting for depression and was not observed among people receiving DMT, suggesting trauma exposure, mood, and inflammatory signalling may be interrelated rather than demonstrating an independent association with MS disease activity [16].

   

In population-based (non-MS) cohorts, violence is associated with comorbidity and multimorbidity through pathways involving direct injury, stress-system dysregulation, and maladaptive coping, which cumulatively elevate allostatic load [18–21]. IPV is also linked to increased smoking, hypertension, hyperlipidaemia, cardiovascular disease, obesity, type 2 diabetes, somatic disorders, chronic pain, reduced engagement with preventative care, lower treatment adherence, and higher rates of polypharmacy and unplanned health service use [19,20,22,23]. These comorbidities are common in MS and associated with poorer outcomes, while disrupted healthcare use may reduce access to treatment and compound the risk of complications [5,24].

### Healthcare response and study rationale

Healthcare settings are often the first or only safe point of contact for individuals affected by violence [25]. In Australia, clinicians have expressed willingness to address violence and contribute to solutions, but inconsistent practice persists due to limited training, role ambiguity, and time constraints [26]. Routine screening has shown some utility in identifying violence, but outside obstetric settings, evidence of benefit for health outcomes remains limited [25,27]. The 2025 United States Preventive Services Task Force (USPSTF) reaffirmed its recommendation to screen for IPV in women of reproductive age, while again finding insufficient evidence for screening in 'vulnerable' and elderly groups [27]. Although no trials reported harms, poorly implemented screening carries hypothetical risks, including distress, re-traumatisation, disengagement from care, and increased time burden [27].

Best practice has shifted toward trauma-informed models that emphasise sensitive inquiry and appropriate response, guided by evidence-based risk indicators [3,25]. In Victoria, Australia, the MARAM (Multi-Agency Risk Assessment and Management) framework is a legislated, system-wide model for FDSV risk assessment and coordinated management, while the World Health Organization's (WHO) LIVES (Listen, Inquire, Validate, Enhance safety, Support access) guides trauma-informed healthcare responses to disclosure [3,25]. Together, they provide complementary approaches for recognition and response. There is little published evidence on how these frameworks function in neurology or whether they require tailoring to account for the relational, clinical, and systemic complexities that shape risk in chronic neurological conditions.

This study addresses these gaps by testing the feasibility of applying best-practice frameworks in specialist neurology clinics, generating prevalence estimates of FDSV in MS, identifying associated psychosocial and clinical factors, and exploring temporal dynamics and associations with comorbidity and outcomes where data allow.

To extend relevance and deepen understanding, the study includes epilepsy and headache comparison cohorts, two chronic, episodic neurological conditions that are often invisible and managed through longitudinal clinical relationships. Like MS, they are shaped by comorbidity, stigma, healthcare complexity, and relational dependence, but carry distinct features that may influence exposure to violence. Early signals of higher prevalence in epilepsy and headache reinforce the value of this design for informing responses across neurology [28–30].

### Objectives

This study aims to:

1. Estimate the prevalence and describe the forms of FDSV experienced by people with MS in specialist neurology care.

2. Identify structural, social, and clinical factors associated with FDSV among people with MS and other neurological conditions to support evidence-based recognition of risk.

3. Investigate current practices in Australian neurology settings for identifying and managing FDSV, including screening, documentation, and referral pathways.

4. Examine systemic barriers and supports for applying evidence-based frameworks such as WHO LIVES, MARAM, and trauma-informed care in neurology settings.

 

5. Prioritise the perspectives of victim-survivors and lived experience contributors to strengthen healthcare responses and inform policy across neurology.

## Primary outcomes

1. Establish the prevalence of FDSV among people with MS attending a multi-site tertiary centre in Victoria with a state-wide consumer base, stratified by demographic, gender, psychosocial, and clinical characteristics.

2. Generate qualitative evidence on current recognition, documentation, and management of FDSV in neurology, identifying barriers and enablers to applying WHO LIVES and MARAM.

## Secondary outcomes

1. Characterise the forms, settings, and perpetrator relationships of FDSV reported by people with MS.

2. Quantify associations between FDSV and demographic, psychosocial, and clinical variables using linked registry, clinical, and survey data, including exploratory temporal analyses and MS outcomes where data permit.

3. Compare MS findings with epilepsy and headache cohorts to distinguish condition-specific and shared patterns.

4. Assess the feasibility of FDSV screening in neurology research, focusing on safe disclosure, safety management, and support pathways, to inform future research and trauma-informed design.

# Methods

This study will run at Alfred Health and Monash University in Melbourne, Victoria, Australia.
   Trial registration: Australian New Zealand Clinical Trials Registry (ANZCTR), ACTRN390279.
   The SPIRIT checklist is provided in S2 File.

## Research team and lived experience collaboration

This study is led by a multidisciplinary team of neurologists, scientists, psychologists, neuropsychologists, nurses, physiotherapists, and social workers. Study design and safety protocols were strengthened through consultation with both internal and external family violence experts, as well as individuals with MS or clinically isolated syndrome (CIS) who are FDSV victim survivors.

   Lived experience contributors with MS or CIS who are victim survivors of FDSV were engaged through existing MSNI consumer and community connections. Those who met the same inclusion criteria for safe participation as study participants contributed their perspectives on how violence intersects with neurological conditions, what factors support safe participation, and how study materials could be made non-triggering and feasible in the context of MS and other neurological conditions. Contributors ensured study materials were safe, affirming, and feasible for people with neurological conditions.

   Consumer advisors with MS who were not victim survivors, engaged through Monash University's volunteer program, focused on feasibility testing. They assessed survey length, clarity of language, and the acceptability of screening scripts to ensure that study procedures were practical and comfortable for participants across the cohort.

   Expert family violence social workers co-developed the screening scripts and refined inclusion and exclusion criteria to minimise risk of harm. They also advised on practical safeguards, including conducting conversations in private, ensuring all participants are routinely asked about safety, avoiding written materials that could be intercepted, and establishing referral pathways within the hospital and to external specialist services.

## Study population and recruitment

Approximately 1,000 people with MS attending the Alfred Hospital MS and Neuroimmunology (MSNI) Clinic will be screened over a 12-month period using convenience sampling during routine appointments. Screening will occur only when individuals are alone and in a private setting, consistent with established clinic procedures for other sensitive research (e.g., reproductive and sexual health).

MS participants who meet inclusion and safety criteria (Table 1) will be invited to participate in quantitative surveys (Arm 3), qualitative interviews (Arm 4a), or both. Approximately 100 participants will be recruited into Arm 3, and at least 20 people with MS into Arm 4a. Purposive sampling will be used to reflect variation in gender, age, lived experience of FDSV, and clinical characteristics across both arms.

Around 100 individuals with epilepsy or headache will be screened to identify at least 12 participants for Arm 4b (qualitative interviews). A hybrid approach of convenience screening and purposive sampling will be used to ensure variation in age, gender, clinical context, and FDSV experience.

In parallel, a minimum of 20 multidisciplinary clinicians involved in MS care (Arm 5a), and at least 12 clinicians from epilepsy and headache services (Arm 5b), will be recruited for qualitative interviews via professional networks and direct contact.

## Sample size

**Quantitative.** As this is a feasibility study, no formal power calculation was undertaken. Quantitative targets were set pragmatically using expert opinion, informed by annual outpatient throughput, the requirement that screening only occur when the participant is private and alone, and the availability of integrated safety and support pathways. The study site reviews approximately 1,000 individual people with MS each year, noting some attend more than once, and recruitment

**Table 1. Inclusion and Exclusion Criteria.**

| Inclusion criteria | Exclusion criteria* |
|---|---|
| People living with MS/Epilepsy/Headache | |
| • History of MS/Epilepsy/Headache<br>• Age 18 or over<br>• Able to give informed consent<br>• Eligible for Medicare | • Non-MS/Epilepsy/Headache diagnosis<br>• Unable to give informed consent<br>• History of violence in the preceding 6 months<br>• Current/recent separation (<6 months)<br>• Ongoing contact with a known perpetrator<br>• Current court/legal/police actions in process<br>• Currently pregnant or breastfeeding<br>• Under the influence of alcohol or illicit substances<br>• Non-English speaking**<br>• Sensory impairment(s) that preclude safe and independent participation***<br>• Mental health comorbidity not stable on treatment<br>• A member from the same family/household has participated in the study already<br>• Declines to be alone and unable to do screening |
| Healthcare professionals (HCPs) | |
| • Age 18 or over<br>• Able to give informed consent<br>• Eligible for Medicare<br>• Practicing in Victoria | • Unable to give informed consent<br>• Vision and/or hearing impairment that precludes safe and independent participation** |

* Exclusions reflect MARAM risk indicators where participation could reasonably increase risk if discovered. All exclusions will be logged to quantify impact on representativeness.

** Interpreters are excluded where use may compromise safety or confidentiality, consistent with Australian evidence on risks in family violence research [37].

*** Individuals with vision or hearing impairments will be excluded only if they cannot participate independently, even with reasonable accommodations.

over 12 months is therefore expected to allow approach of approximately 1,000 Victorian residents with MS for screening when privacy can be assured.

Survey completion is expected in approximately 100 participants, reflecting anticipated uptake for a sensitive topic, with recruitment continuing within the recruitment window if higher numbers are feasible. An estimated range of 70–150 completed surveys is anticipated within the recruitment window, depending on uptake and safety considerations.

To contextualise feasibility estimates, screening $n \approx 1,000$ supports estimation of key feasibility proportions (for example uptake, privacy achievable, and support pathway actions) with an approximate 95% CI half width up to ±3.1%. Survey completion $n \approx 100$ supports the same feasibility proportions with an approximate 95% CI half width up to ±9.8%.

**Qualitative sample size and analytic sufficiency.** Qualitative interview targets are flexible and guided by thematic saturation and analytic sufficiency. In this study, thematic saturation is defined as the point at which additional interviews do not generate new substantive codes or concepts that modify the coding framework or alter interpretation of themes relevant to the study aims, including domains aligned with MARAM, LIVES, and condition specific features. Saturation will be assessed iteratively within each participant group during data collection and analysis. An audit trail will be maintained that includes dated versions of the coding framework, analytic memos, and analysis meeting notes documenting when codes are added, merged, or refined and the rationale for these changes. A minimum of approximately 30 interviews will be undertaken to support depth and diversity of perspectives across participant groups and key domains. Additional interviews will be undertaken only if the audit trail indicates continued emergence of new substantive concepts or insufficient representation across key domains required to address the study aims.

## Study status and projected timeline

The study is expected to be completed within 3 years of commencement. Recruitment for Arm 5, which includes healthcare professional interviews, commenced in September 2025 and is expected to be completed in the second quarter of 2026. Arms 1–4 and Arm 6 have not yet commenced. Screening of people with MS for Arm 1 is planned to begin in the first quarter of 2026 and is expected to continue for approximately twelve months. Survey recruitment for Arm 3 and qualitative interviews for Arms 4a and 4b are expected to run for a further six months following completion of screening which is projected for the first and second quarters of 2027. Retrospective data linkage for Arm 2 will occur during the screening and survey period. No study results have been generated. Triangulation for Arm 6 will commence once data collection is complete and is expected to be completed by the end of 2027. Results are expected within approximately six months after completion of data collection for each arm. The schedule of enrolment, interventions and assessments and study flow is provided in Figs 1 and 2, eligibility criteria are detailed in Table 1.

## Research design

This study employs a convergent mixed-methods design to integrate quantitative and qualitative data to inform the adaptation of best-practice frameworks within neurological care. While centred on MS, the inclusion of epilepsy and headache enables condition-level comparisons across disorders with episodic trajectories, variable age of onset, and predominantly invisible symptoms. These features shape healthcare engagement and may influence how clinicians recognise and respond to FDSV. Including these conditions facilitates a nuanced examination of how violence intersects with neurological processes and condition-specific profiles.

Six complementary study arms will run concurrently to investigate prevalence, clinical response, lived experience, and system-level barriers and enablers (Fig 3; Table 2). Quantitative arms include screening data (Arm 1), retrospective registry and EMR data (Arm 2), and validated surveys with people with MS (Arm 3). Qualitative arms comprise semi-structured interviews with people with MS (Arm 4a), individuals with epilepsy or headache (Arm 4b), and clinicians involved in MS and subspecialty headache and epilepsy care (Arms 5a and 5b). Data will be analysed in parallel and integrated through

| Activity | t-1<br>HCP arm | t0<br>Screening | t1<br>Enrolment | t2<br>Surveys | t3<br>Interviews | t4<br>Linkage | t5<br>Integration |
|---|---|---|---|---|---|---|---|
| **Enrolment** | | | | | | | |
| Eligibility and private safety check | | X | | | | | |
| Explain study, opt-in for screening + data linkage | | X | X | | | | |
| Assess inclusion and MARAM safety | | X | X | | | | |
| Consent to surveys or interviews | | | X | | X | | |
| **Procedures** | | | | | | | |
| FDSV screen by clinician (Arm 1) | | X | | | | | |
| Safety support and referral as needed | | X | X | X | X | X | X |
| Allocate to arms based on eligibility + consent | | | X | | | | |
| MS surveys (Arm 3) | | | | X | | | |
| Interviews MS (Arm 4a) | | | | | X | | |
| Interviews epilepsy/headache (Arm 4b) | | | | | X | | |
| Clinician interviews MS (Arm 5a) | X | | | | | | |
| Clinician interviews epilepsy/headache (Arm 5b) | X | | | | X | | |
| Optional member checking | X | | | | X | | X |
| **Assessments** | | | | | | | |
| Brief FDSV risk screen | | X | | | | | |
| Sociodemographic and psychosocial data | | | | X | X | X | |
| MS disease and DMT data (MSBase + records) | | | | | | X | |
| Comorbidities + cognition from records | | | | | | X | |
| Survey outcomes: trauma, mental health, violence history (incl. disability targeted and tech facilitated), substance use, self-esteem, fatigue, QoL, head injury | | | | X | | | |
| Qualitative lived experience data (MS) | | | | | X | | |
| Qualitative lived experience data (epilepsy/headache) | | | | | X | | |
| Clinician perspectives on FDSV response | | | | | X | | |
| Link screening, survey, clinical datasets | | | | | | X | |
| Analytic datasets for prevalence, associations | | | | | | X | X |
| Mixed methods triangulation (Arm 6) | | | | | | | X |
| **Safety** | | | | | | | |
| Adverse event monitoring | | X | X | X | X | X | X |
| Pause, reschedule, withdraw | | X | X | X | X | | |

**Fig 1. Schedule of enrolment, procedures, assessments, and safety monitoring across study timepoints.** Columns indicate t minus 1 (HCP arm), t0 (screening), t1 (enrolment), t2 (surveys), t3 (interviews), t4 (linkage), and t5 (integration). Rows are grouped as enrolment activities, study procedures, assessments, and safety processes. "X" indicates when each activity occurs.

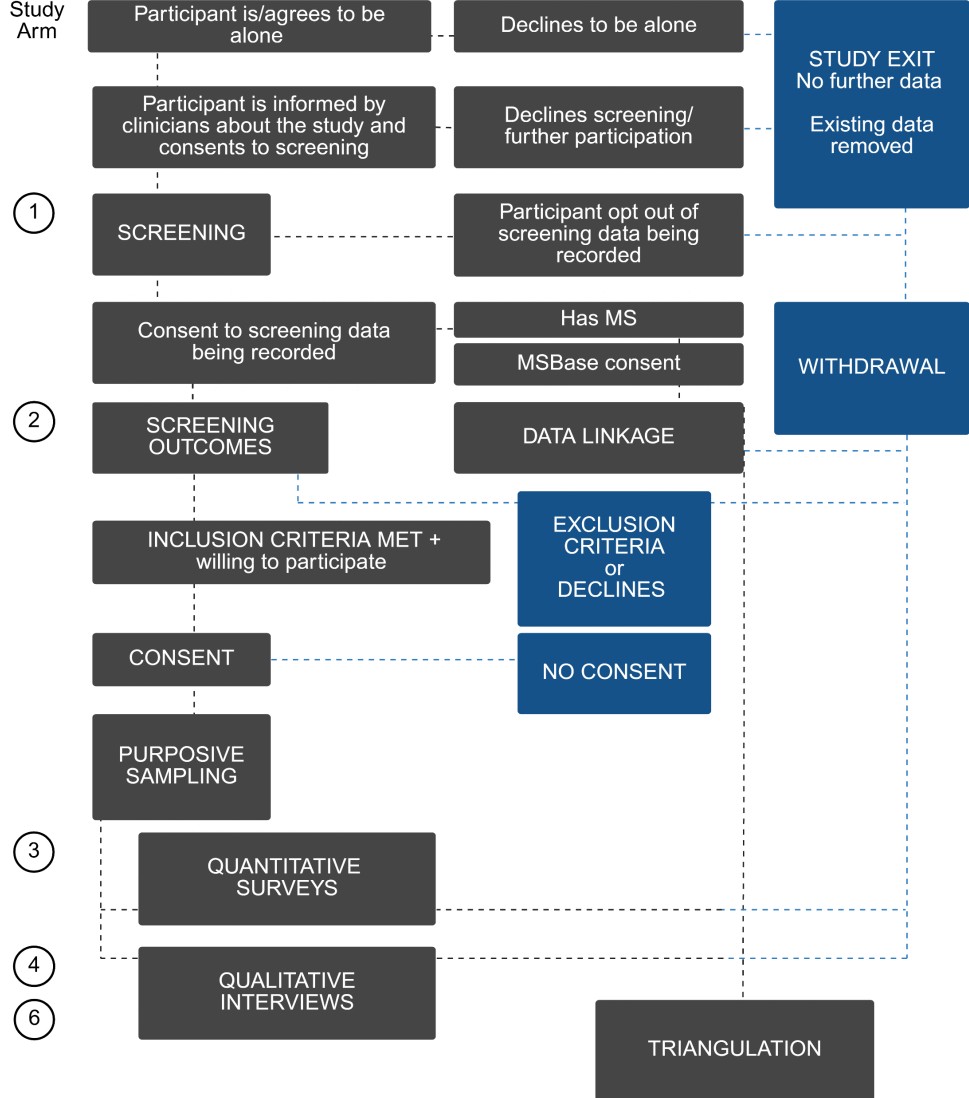

**Fig 2. Participant flow and consent pathway across study arms.** Following a private safety check, participants may proceed to clinician screening (Arm 1) or decline (study exit). Screening outcomes can be recorded for linkage (Arm 2) for those with MS and MSBase consent, followed by eligibility assessment and consent for further participation. Eligible participants may complete quantitative surveys (Arm 3) and or qualitative interviews (Arm 4). Mixed-methods integration occurs through triangulation (Arm 6). Boxes indicating exit, exclusion, no consent, or withdrawal represent non progression points.

triangulation (Arm 6) to generate a comprehensive understanding of FDSV in neurology, encompassing prevalence, risk patterns, clinical response, and structural dynamics.

The study is designed not only to inform implementation through evidence and feasibility testing but also centre the voices of people with lived experience and ensure their insights inform clinical and systemic response.

## Consent

**Consent and clinician led screening for Arms 1–2.** Screening and verbal opt in consent for data linkage will be obtained by the treating clinician within routine care, and only when the participant is alone and privacy can be assured.

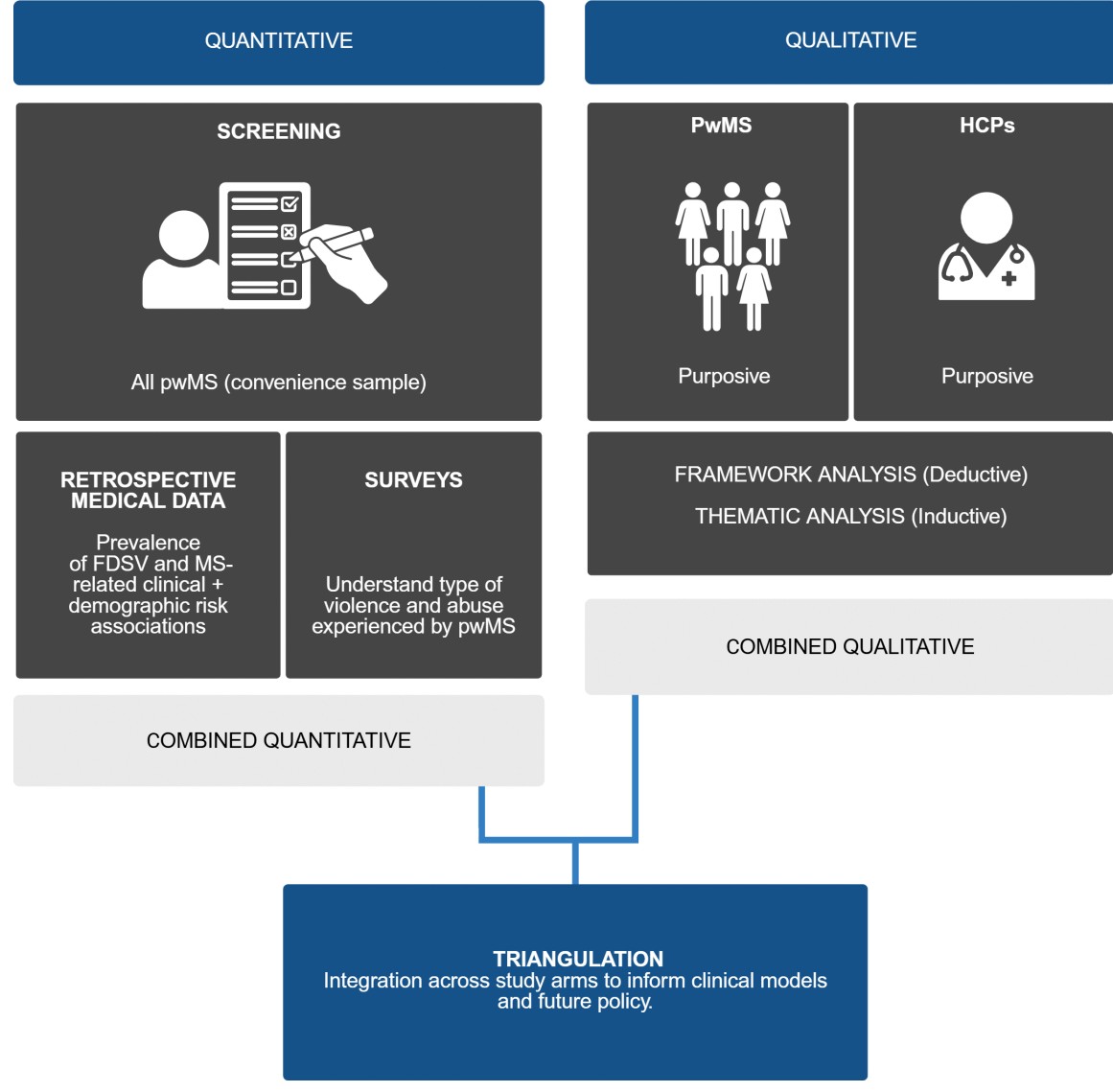

**Fig 3. Convergent mixed-methods framework for MS focused FDSV research.** Quantitative components comprise screening of pwMS, retrospective medical data linkage for prevalence and MS related clinical and demographic associations, and surveys characterising forms of violence and abuse. Qualitative components comprise purposive interviews with pwMS and people with headache and epilepsy, and with HCPs, analysed using a combined deductive MARAM and LIVES framework and inductive thematic analysis. Findings are integrated through triangulation across study arms to inform clinical models and future policy.

This approach aligns with WHO ethical and safety recommendations and related guidance that prevalence research should use methods that minimise underreporting, and that willingness to disclose is influenced by the interactional context of enquiry, including rapport and trust [31–33]. Clinician led screening within an established therapeutic relationship is therefore intended to support safer disclosure, reduce predictable under ascertainment, and enable immediate duty of care and activation of existing support pathways when risk is identified [31].

Verbal consent is used to minimise participation visibility and reduce the risk of written or electronic study materials being discovered in the context of surveillance or coercive control.

**Table 2. Convergent mixed methods design.**

| Research Type | Study Arm | Data | Purpose/Outcome |
|---|---|---|---|
| Quantitative Data | 1. Screening | Screening data | All people with MS or CIS attending a MS clinic will be screened for lived experience of FDSV. Screening will also review study inclusion and exclusion criteria, assess safety, and identify the most appropriate setting for participation. |
| | 2. Retrospective Data | EMR MSBase Linkage with screening data | Explore the intersection of MS and FDSV by identifying associated factors, characterising reported types, and, where data permit, examining temporal dynamics and associations with MS and health outcomes using descriptive statistics and multivariable methods such as IPTW or regression. Draw on screening, registry, clinical, and survey data to identify patterns that may inform risk recognition and tailored responses in MS care. |
| | 3. Surveys | Survey responses | |
| Qualitative Data | 4A. (people with MS) 4B. Interviews (people with epilepsy or headache) | Semi-structured interviews | Examine risk recognition, screening, and response practices from with neurological condition perspective and compare these with MARAM/LIVES frameworks. |
| | | | Conduct descriptive content and secondary thematic analysis of lived experiences and their intersection with MS. |
| | 5A. Interviews (MS HCPs) 5B. Interviews (epilepsy and headache HCPs) | Semi-structured interviews | Understand current practices for risk recognition, screening, and response, including barriers and enablers to MARAM/LIVES implementation in MS and neurology care. |
| Convergent Mixed Methods | 6. Triangulation | Combined quantitative and qualitative data | Integrate qualitative and quantitative findings to compare MS with other episodic neurological conditions, examine differences across disciplines, practice settings, and demographics, and identify convergent and divergent patterns. Use these insights to refine prevalence estimates, clarify risk factors, and develop evidence-based recommendations for MS- and neurology-tailored screening and response, with implications for future policy. |

To minimise perceived influence, clinicians will use a standardised script emphasising that participation is voluntary and declining will not affect clinical care. Participants may skip any question and may pause, stop, or defer at any time. Where feasible, participants may choose to complete consent processes with a trained member of the research team who is not involved in their clinical care. If distress or safety concerns arise, screening will be paused or discontinued and the pre-specified distress and safety pathway activated.

Registry linkage (Arm 2) will only be undertaken for individuals who have previously consented to inclusion in MSBase [34].

**Consent for Arms 3–5.** Written or electronic consent for participation in surveys and interviews in Arms 3–5 will be obtained by a trained member of the research team in accordance with local procedures. Participants in Arms 3–5 will also be given the option to provide extended consent for the use of their coded data in future related research. (Fig 2)

## Quantitative data: Study Arms (1–3)

**Arm 1: Participant screening.** The co-designed screening script defines FDSV, explains voluntary participation, and assesses lived experience and eligibility for Arms 3 and 4 using MARAM-aligned risk indicators [3] (Table 1). It includes prompts on technology-facilitated violence, determines whether participation should occur onsite, remotely or not at all, and outlines support pathways. If screening cannot occur because the participant cannot be seen alone, privacy cannot be assured, or clinical priorities preclude screening, the reason will be recorded.

Arm 1 will capture predefined feasibility indicators covering reach and uptake, safe disclosure setting, and safety management and support pathway actions, for descriptive reporting

**Arm 2: Retrospective clinical and registry data.** Screening responses will be linked with MSBase and clinical records to examine associations between FDSV and neurological, demographic and psychosocial factors relevant to MS

care. MSBase will provide data on disease onset, course, DMTs, relapse history, and longitudinal expanded disability status scale scores. Clinical records will contribute demographic information including sex assigned at birth, gender identity, and geographical location, as well as comorbidities and neuropsychology cognitive assessments where available. Analysis of the linked dataset will strengthen risk recognition by identifying MS related patterns associated with FDSV and informing response strategies tailored to neurology.

**Arm 3: Bio-psychometric surveys.** A minimum of 100 participants with MS will complete validated bio-psychometric instruments, selected in consultation with family violence specialists to align with study objectives while minimising cognitive and time burden. These measures (Table 3) capture additional demographic data including sexual orientation, relationship history, cultural background, and education level, which may be mapped to an approximate years of education variable for analysis where required, as well as trauma exposure, mental health symptoms, violence history (including disability-targeted and technology-facilitated violence), substance use, self-esteem, and head injury, alongside MS-specific indices of fatigue and quality of life. Fatigue is deliberately included given its prominence as one of the most common, disabling, and clinically significant symptoms in MS [35,36].

**Feasibility, safe disclosure, and response pathways.** Feasibility of FDSV screening in neurology research will be evaluated descriptively using predefined indicators, with denominators reported for each measure.

Screening process

- Proportion of eligible encounters where privacy can be assured

- Uptake and completion once commenced

- Decline and deferral, with reasons where recorded

- Noncompletion because privacy or safety cannot be assured

- Exclusions under the pre-specified safety criteria in Table 1, with reasons summarised where recorded

Safe disclosure

**Table 3. Survey Instruments.**

| Instrument | Domain Assessed | Rationale |
| --- | --- | --- |
| MS-QOL [38] | MS-specific quality of life | Captures physical and mental health impact of MS |
| MFIS [39] | Fatigue | Common MS symptom influencing function and wellbeing |
| Composite Abuse Scale (CAS) [40] | Interpersonal violence | Assesses physical, emotional, and sexual violence |
| Abbreviated AAS – Disability [41] | Disability-targeted violence | Screens for violence specific to disability context |
| TAR scale [42] | Technology-facilitated violence | Captures control, surveillance and online violence |
| DASS-21 [43] | Depression, anxiety, and stress | Screens for psychological distress |
| ACE [44] | Adverse Childhood Events | Risk factors for adult health and violence exposure |
| RSES [45] | Self-esteem | Captures psychosocial resilience and coping |
| ASSIST-Lite [46] | Substance use | WHO-endorsed screen for alcohol and drug use |
| Abbreviated IPV Head Injury Assessment [47] | Traumatic brain injury | Screens for IPV-related head trauma |

Surveys were selected for their prior validation, particularly in Australian research, and in consultation with family violence experts to capture broad data while keeping survey length and burden manageable.

MS-QOL = Multiple Sclerosis Quality of Life scale, MFIS = Modified Fatigue Impact Scale, CAS = Composite Abuse Scale, AAS = Abuse Assessment Screen, TAR = Technology-facilitated Abuse Recognition measure, DASS-21 = Depression, Anxiety and Stress Scale (21 items), ACE = Adverse Childhood Experiences, RSES = Rosenberg Self-Esteem Scale, ASSIST-Lite = Alcohol, Smoking and Substance Involvement Screening Test (Lite version), IPV = Intimate Partner Violence.

- Participation setting (on site, remote, not appropriate)

- Screening paused or discontinued due to distress or safety concerns

  Screening harms

- Any distress or safety issue during or immediately following screening that triggers the distress protocol, results in discontinuation, or requires escalation under institutional pathways

  Safety management and support pathways

- Resources or information provided

- Referral offered, accepted, and initiated

- Follow up requested and completed

- Number and type of actions initiated following screening

**Quantitative data analysis.**  Descriptive analyses will estimate the prevalence and examine variation in experiences of family, domestic, and sexual violence according to demographic, psychosocial, and clinical characteristics, including disability severity, cognitive function, comorbidities, and MS clinical variables.

Patterns of missing data will be reported by study component and variable, with denominators provided for all estimates, and sensitivity analyses will be conducted where appropriate. Missingness for routinely captured MS variables is anticipated to be low based on established site recording practices, whereas missingness for survey items is uncertain given the sensitive content and the option to skip questions. No imputation will be undertaken. The extent and pattern of missing or skipped survey responses, including non-response to specific questions, will be interpreted as a feasibility and acceptability finding. As this is a pilot study, the sample size and the number of participants with disclosed lived experience are uncertain. Where sufficient data are available, further analyses will be undertaken to explore temporal dynamics and associations with MS outcomes, using multivariable regression or inverse probability of treatment weighting methods as appropriate.

## Qualitative data: Study arms (4–5)

**Arm 4a: Interviews with people with MS.**  Semi-structured interviews will explore the lived experience of FDSV among people with MS, focusing on disclosure, healthcare interactions, and the barriers and enablers to help-seeking within clinical neurology settings. Interviews will also examine perceptions of neurology's role in recognising and responding to violence, the ways MS may shape patterns of violence and control, and perceived barriers and enablers to integrating best-practice frameworks into care. Attention will be given to MS-specific features that influence these experiences.

Interview guides were co-developed by the multidisciplinary research team, structured around current practice, perspectives, and alignment with MARAM and WHO LIVES, and designed for consistent use across all participant groups (people with neurological conditions and healthcare professionals).

**Arm 4b: Interviews with people with epilepsy and headache.**  Using the same interview guide as Arm 4a, interviews will be conducted with individuals living with epilepsy or headache to enable comparison across subspecialties. This parallel approach will support the identification of both shared and condition-specific factors influencing FDSV experiences and clinical interactions, enhancing the relevance of findings across neurology.

**Arm 5a: Interviews with MS healthcare professionals.**  Multidisciplinary clinicians involved in MS care will be interviewed in person or via secure videoconferencing. Semi-structured interviews will explore current approaches to recognising and responding to FDSV, including personal and system-level barriers and enablers. Clinicians will also

be asked about their perceptions of how neurological conditions such as MS intersect with experiences of FDSV. Interview guides will include structured items on clinicians' knowledge, prior training, and perspectives on neurology's role in recognising and responding to violence. Responses to these items will form a small quantitative component reported descriptively, alongside qualitative analysis of interview transcripts using a framework and thematic approach.

**Arm 5b: Interviews with healthcare professionals in epilepsy and headache.** Clinicians working in epilepsy and headache subspecialty settings will be interviewed using the same framework as Arm 5a. Comparative analysis will examine how FDSV is recognised and addressed across different areas of neurology, identifying shared barriers, challenges, and enablers, as well as subspecialty-specific considerations. This will support opportunities for cross-learning and inform improvements across the broader neurology workforce.

**Qualitative data analysis.** This study adopts a constructivist epistemological stance, recognising that meaning is co-produced through interaction between participants and researchers, with reflexive attention to how researcher perspectives and disciplinary backgrounds shape interpretation. A combined inductive and deductive analytic approach will be used. Deductive framework analysis will apply the MARAM and LIVES domains, with matrix charting used to systematically compare responses across participant groups and clinical contexts (Fig 4). In parallel, inductive thematic analysis will identify emergent concepts that extend beyond the pre-defined frameworks, with interpretation reviewed collaboratively by the multidisciplinary research team.

To support analytic consistency across the multidisciplinary team, an initial subset of transcripts will be coded collaboratively by at least two researchers, with coding decisions discussed in analysis meetings. This calibration phase will be used to refine the coding framework, agree definitions and boundaries for key codes, and document decision rules for applying the MARAM and LIVES domains alongside inductive concepts. Dated versions of the coding framework will be retained, supported by analytic memos and meeting notes that record when codes are added, merged, or refined and why.

NVivo will be used to support data management and coding. Rigour will be supported through documentation of analytic decisions and team-based discussion during coding and interpretation.

Findings will be integrated with quantitative data through triangulation (Arm 6), contributing to a comprehensive and multidimensional understanding of how FDSV is experienced, recognised, and addressed in neurology care.

## Mixed methods convergent design

**Arm 6: Triangulation.** This arm integrates findings from all study components to generate a comprehensive understanding of FDSV in MS and across neurology. Quantitative and qualitative data will be analysed separately but in parallel within a convergent mixed methods model. Findings from each study arm will be organised into common domains, and integration will be achieved through triangulation matrices and side-by-side comparison. This process will enable identification of convergent, divergent and complementary patterns in individual experience, clinical presentation, and service context. Analyses will compare participants with and without lived experience of FDSV, across gender, condition, care setting, discipline type, and location.

Triangulation will support the identification of condition-specific, disability-related, and structural factors influencing recognition, response, and support.

## Data collection and management

Data will be collected through MSBase, REDCap and clinical records and will be linked using unique study codes. Identifiable information will be stored separately from research data in secure and access-controlled systems in accordance with institutional requirements. Only authorised members of the research team involved in recruitment and safety procedures will have access to identifiable information.

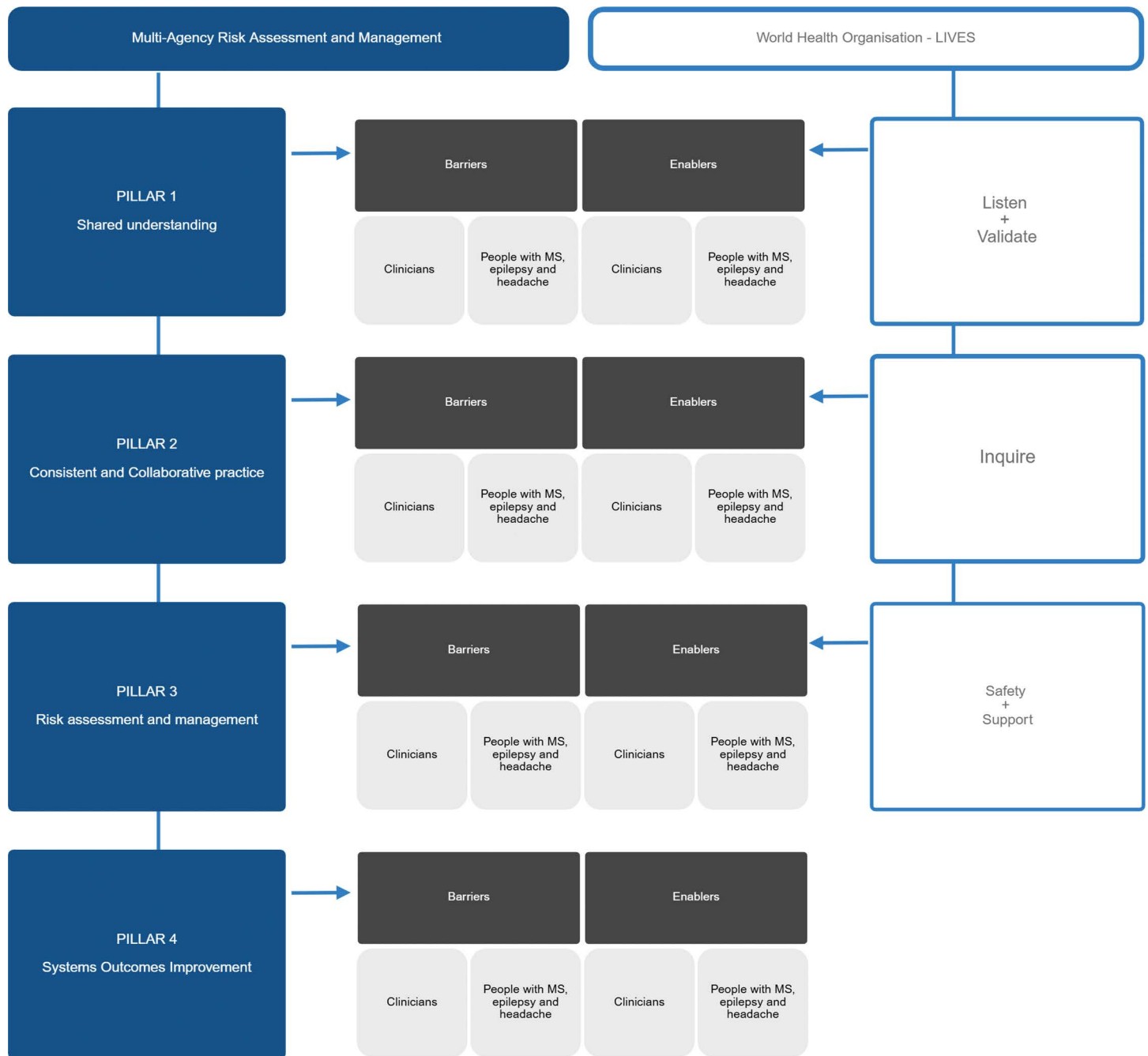

**Fig 4. Deductive analytic framework aligning MARAM pillars with LIVES domains to structure qualitative analysis of barriers and enablers.** Pillars 1 to 3 (shared understanding; consistent and collaborative practice; risk assessment and management) are examined for barriers and enablers reported by HCP and participants, mapped to LIVES elements (listen and validate; inquire; safety and support). Pillar 4 (systems outcomes improvement) is included as the systems level domain informing implementation and improvement outputs.

Interviews will be audio recorded and professionally transcribed. Transcripts will be reviewed and coded to remove direct identifiers before analysis. Survey responses and interview transcripts will be stored in REDCap under the assigned study codes. MSBase and clinical data will be extracted in coded form for analysis.

Coded data will be used for all quantitative and qualitative analyses and will not contain direct identifiers. Re identification will occur only where required for participant safety, such as adverse event follow up, and will be undertaken solely by authorised clinical researchers following institutional protocols.

Participants may withdraw at any time before data aggregation. If a participant withdraws, all identifiable and coded data relating to that individual will be removed. All data management and retention processes will comply with institutional and national guidelines.

### Ethics

This study was approved by the Alfred Health Human Research Ethics Committee (project number: 115129; approval date: 16 June 2025) and the Monash University Human Research Ethics Committee (project number: 48580; approval date: 23 July 2025) with commencement in August 2025. The methodology is informed by World Health Organization guidance and established ethical principles for FDSV research [32]. Safety protocols were co-developed with family violence experts, and victim survivors with MS informed design, language, and risk procedures.

Quantitative reporting will follow STROBE, qualitative reporting COREQ, and mixed-methods reporting GRAMMS. Research on FDSV in people with MS and other neurological conditions presents distinct ethical challenges, including risks of distress, disengagement, inadvertent disclosure, or escalation of harm if participation is discovered. Key risks and mitigation strategies are summarised in S1 Table. HREC approval letters are provided in S3 and S4 Files.

Harms in this study refer to distress or safety concerns that arise during screening, surveys or interviews. These will be assessed non systematically through direct observation by the treating clinician or research team member during participant interactions. Any indication of distress or safety risk will be managed in accordance with the study distress protocols and local procedures, with immediate support and referral where required. Study harms and any adverse events requiring escalation will be managed through established institutional pathways and reported to the approving ethics committees in line with institutional requirements. The approving HREC will monitor any reportable events in accordance with its standard oversight processes.

### Dissemination

Findings will be disseminated through peer-reviewed publications, academic presentations, and the study's online platform for participants. Results will inform prevalence reporting, approaches to screening, and the integration of trauma-informed frameworks into neurology. Inclusion of epilepsy and headache will enable cross-subspecialty comparison and support broader application across neurology.

### Consumer and public involvement

Victim-survivors with multiple sclerosis or clinically isolated syndrome contributed to the design of this study, including refinement of research questions, language, safety protocols, screening tools, and interview guides. Consumer advisors with MS who were not victim-survivors were involved in feasibility testing of survey length, clarity, and acceptability. Family violence social workers with lived experience perspectives co-developed the screening scripts and informed inclusion and exclusion criteria to minimise risk. Lived experience contributors will continue to be involved through data analysis and interpretation, manuscript preparation, and dissemination of findings.

## Box 1. Definition and Scope of Family, Domestic, and Sexual Violence (FDSV)

**Definitions (Victoria Australia):**

FDSV is defined as any behaviour (past or present) that controls, dominates, or causes a family member to fear for their safety or wellbeing, or that of another person. Throughout this project we use the WHO definition of violence, [40, 41] noting that "abuse" is used interchangeably across health, legal and policy contexts.

**Forms of FDSV:**

- Emotional or psychological violence (e.g., intimidation, manipulation, humiliation)
- Economic violence (e.g., financial control or restriction of access to funds)
- Sexual violence (e.g., coercion, assault)
- Social isolation (e.g., limiting movement or contact)
- Reproductive violence (e.g., control of contraception or pregnancy)
- Spiritual violence (e.g., restricting or demeaning religious or cultural practice)
- Technology-facilitated violence (e.g., surveillance, digital harassment)
- Exploitation of disability (e.g., using disability to obtain resources or assert control)
- Care-related violence (e.g., restricting access to medical care or services, misuse of finances, psychological violence related to disability, neglect or coercion in care provision)

**Contexts of Occurrence:**

- Intimate partner violence (including current, former, or brief relationships)
- Family and kinship networks (family members, close family friends, elders, or religious ties)
- Care relationships (where a paid or unpaid carer uses their role to harm, control, or neglect)
- Young people can also use violence or be victims of violence within their family.

**Revictimisation:** experiencing violence on more than one occasion, across different relationships, or life stages.

**MARAM Framework:** Victoria's Multi-Agency Risk Assessment and Management Framework, a state-wide structure for consistent identification, assessment, and management of FDSV across services.

**WHO LIVES Framework:** World Health Organization guidance for healthcare responses to violence, built around five steps: Listen, Inquire, Validate, Enhance safety, and Support access.

**Trauma-informed care:** A strengths-based framework for service delivery that acknowledges the widespread impact of trauma and understands how to respond to it

## Supporting information

**S1 Table. Study risks and mitigation plan.**
(DOCX)

**S2 File. SPIRIT checklist.**
(DOCX)

**S3 File. Alfred Health Human Research Ethics Committee approval.**
(PDF)

**S4 File. Monash University Human Research Ethics Committee approval.**
(PDF)

## Acknowledgments

We thank the Alfred Health Family Violence Program (Melbourne, Victoria) and regional social worker Vida Luimaite (Geelong Victoria), for their guidance in ensuring MARAM-aligned safe participation in this study.

## Author contributions

**Conceptualization:** Cassie Nesbitt, Genevieve Rayner, Kristen Timmens, Helmut Butzkueven, Anneke van der Walt, Vilija G. Jokubaitis.

**Data curation:** Cassie Nesbitt, Christina Kazzi, Rubina Alpitsis, Emma Foster, Jason Ray, Helmut Butzkueven, Anneke van der Walt, Vilija G. Jokubaitis.

**Formal analysis:** Cassie Nesbitt, Kristen Timmens, Christina Kazzi, Rubina Alpitsis, Emma Foster, Jason Ray, Joanne Crosby, Helmut Butzkueven, Anneke van der Walt, Vilija G. Jokubaitis.

**Investigation:** Cassie Nesbitt, Helmut Butzkueven, Anneke van der Walt, Vilija G. Jokubaitis.

**Methodology:** Cassie Nesbitt, Genevieve Rayner, Kristen Timmens, Christina Kazzi, Rubina Alpitsis, Joanne Crosby, Helmut Butzkueven, Anneke van der Walt, Vilija G. Jokubaitis.

**Project administration:** Cassie Nesbitt, Anneke van der Walt, Vilija G. Jokubaitis.

**Resources:** Cassie Nesbitt, Genevieve Rayner, Rubina Alpitsis, Joanne Crosby, Helmut Butzkueven, Anneke van der Walt, Vilija G. Jokubaitis.

**Supervision:** Genevieve Rayner, Rubina Alpitsis, Joanne Crosby, Helmut Butzkueven, Anneke van der Walt, Vilija G. Jokubaitis.

**Validation:** Cassie Nesbitt.

**Visualization:** Cassie Nesbitt.

**Writing – original draft:** Cassie Nesbitt, Genevieve Rayner, Kristen Timmens, Christina Kazzi, Rubina Alpitsis, Emma Foster, Jason Ray, Joanne Crosby, Helmut Butzkueven, Anneke van der Walt, Vilija G. Jokubaitis.

**Writing – review & editing:** Cassie Nesbitt, Genevieve Rayner, Kristen Timmens, Christina Kazzi, Rubina Alpitsis, Emma Foster, Jason Ray, Joanne Crosby, Helmut Butzkueven, Anneke van der Walt, Vilija G. Jokubaitis.

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
