## [Decision Letter · Decision Letter 0]

5 Feb 2026

PLOS One

Dear Dr. Nesbitt,

Thank you for submitting your manuscript to PLOS ONE. After careful consideration, we feel that it has merit but does not fully meet PLOS ONE’s publication criteria as it currently stands. Therefore, we invite you to submit a revised version of the manuscript that addresses the points raised during the review process.

The reviewers commend the importance of your study topic, the comprehensive literature review, and the robust mixed-methods design, as well as the meaningful involvement of victim-survivors in co-development. They agree that your study aims are relevant and achievable and that the methodology is appropriate for addressing this important issue.

However, both reviewers have identified several points that require minor revisions to strengthen the manuscript. Please address the following comments in your revised submission:

Clarify the informed consent process, particularly addressing any concerns about potential influence when screening and consent are obtained by treating clinicians, given the sensitive nature of family violence.Specify whether data on sexual orientation, disease-modifying therapy use, and years of education will be collected, and if so, how these variables will be incorporated.Provide greater detail on how the feasibility of family domestic and sexual violence (FDSV) screening in neurology research will be evaluated within your data analysis plan, including aspects of safe disclosure, safety management, and support pathways.Elaborate on the calculation of the sample size in the quantitative arm of the study.Describe how thematic saturation will be assessed in the qualitative arm, including details on how "thematic saturation" is defined and calculated. Clarify the rationale for the target number of approximately 100 surveys, and specify any estimated range (minimum and maximum) for survey numbers.Outline your approach to managing missing or incomplete data, considering the potential for a substantial amount of such data in this extensive study.Expand the Background/Context section to include results from other studies worldwide on intimate partner violence (IPV) and family domestic and sexual violence (FDSV) among people with multiple sclerosis, to further support the research rationale.

We look forward to receiving your revised manuscript.

Kind regards,

Yordanis Enríquez Canto, Ph.D.

Academic Editor

PLOS One

2. In the online submission form you indicate that your data is not available for proprietary reasons and have provided a contact point for accessing this data. Please note that your current contact point is a co-author on this manuscript. According to our Data Policy, the contact point must not be an author on the manuscript and must be an institutional contact, ideally not an individual. Please revise your data statement to a non-author institutional point of contact, such as a data access or ethics committee, and send this to us via return email. Please also include contact information for the third party organization, and please include the full citation of where the data can be found.

Reviewers' comments:

Reviewer's Responses to Questions

**Comments to the Author**

1. Does the manuscript provide a valid rationale for the proposed study, with clearly identified and justified research questions?

Reviewer #1: Yes

Reviewer #2: Yes

2. Is the protocol technically sound and planned in a manner that will lead to a meaningful outcome and allow testing the stated hypotheses?

Reviewer #1: Yes

Reviewer #2: Yes

3. Is the methodology feasible and described in sufficient detail to allow the work to be replicable?

Reviewer #1: Yes

Reviewer #2: Yes

4. Have the authors described where all data underlying the findings will be made available when the study is complete?

Reviewer #1: Yes

Reviewer #2: Yes

5. Is the manuscript presented in an intelligible fashion and written in standard English?

Reviewer #1: Yes

Reviewer #2: Yes

You may also provide optional suggestions and comments to authors that they might find helpful in planning their study.

Reviewer #1: Dear Dr. Nesbitt and colleagues,

Thank you for submitting this manuscript titled, “Study protocol for a mixed methods convergent investigation of family domestic and sexual violence in multiple sclerosis and broader neurology in Australia”. Family violence is an under-studied, but important topic in people with multiple sclerosis, emphasizing the importance of this study protocol. There are several strengths of this manuscript, including the co-development with victim-survivors with MS, the mixed-methods design, and the comprehensive literature review. The study aims are relevant and achievable. The methodology is appropriate for the study aims and potential results will have vital ramifications for the care of people with MS. I have a few minor points to improve the manuscript.

1) The authors note that screening and consent will be obtained by the treating clinician. Is there a concern that this could influence the informed consent process? This is particularly relevant with regard to a study focused on family violence.

2) Would data be gathered about sexual orientation, disease-modifying therapy use, or years of education?

3) One of the study secondary outcomes is to “Assess the feasibility of FDSV screening in neurology research, focusing on safe disclosure, safety management, and support pathways, to inform future research and trauma-informed design.” However, it is unclear how this will be evaluated in the data analysis plan. Would you be able to provide clarity?

Thank you for submitting this protocol, which may support other researchers/clinicians working in this area.

Reviewer #2: This is a generally well-written and methodologically sound research protocol that addresses a fundamental, yet overlooked issue. The overall methodological foundation is strong. Here are some issues that need to be addressed:

1) While your argument about power analysis is correct, the authors should explain how they calculated the sample size in the quantitative arm

2) In the qualitative arm, how do the authors assess the saturation? Please provide details on how "thematic saturation" is calculated. Also, it is mentioned that around 100 surveys will be performed; what is the basis of this number? And also, is there an estimated target range for the number of surveys (minimum and maximum)?

3) Due to the extent of the study, it is possible that the authors might encounter a large number of missing/incomplete data. How will the authors handle the missing/incomplete data?

4) Please provide results of other studies worldwide on IPV and FDSV among MS patients in the Background/Context section to further support your research rationale.

**Do you want your identity to be public for this peer review?** For information about this choice, including consent withdrawal, please see our Privacy Policy

Reviewer #1: No

Reviewer #2: No

---

## [Author Response · Author response to Decision Letter 1]

19 Feb 2026

Response to reviewers

General comment

Reviewer comment: The reviewers commend the importance of the study topic, the comprehensive literature review, the robust mixed methods design, and the meaningful involvement of victim survivors in co development.

Response: We thank the reviewers for the encouraging feedback. We particularly appreciate acknowledgement of the victim survivors with MS on the research team, whose lived expertise has directly informed the co design and conduct of the research. Their courage and determination to use lived experience to drive positive change for others merits explicit recognition, and we are grateful the reviewers took the time to highlight this.

Location: Not applicable.

Comment 1

Reviewer comment: Clarify the informed consent process, particularly addressing any concerns about potential influence when screening and consent are obtained by treating clinicians, given the sensitive nature of family violence.

Response: The Consent section has been expanded to clarify clinician led screening and verbal opt in consent for Arms 1 and 2, including safeguards to minimise perceived influence. Treating clinicians use a standardised script that states participation is voluntary and will not affect care. Participants may skip any question and may pause, stop, or defer at any time. Where feasible, and where preferred by the participant, consent may be completed with a trained research team member who is not involved in clinical care.

Location: Consent, pages 15 to 16.

Comment 2

Reviewer comment: Specify whether data on sexual orientation, disease modifying therapy use, and years of education will be collected, and if so, how these variables will be incorporated.

Response: Sexual orientation and education level will be collected in Arm 3 within survey demographics. Education will be recorded as highest level attained, which serves as a proxy for years of education in the Australian context. Sexual orientation and education are not routinely captured in clinical records or MSBase and will therefore be incorporated in Arm 3 survey based analyses, not the linked Arm 2 dataset. Disease modifying therapy use will be captured from MSBase and clinical records as part of routine MS history and incorporated in linked Arm 2 analyses alongside other MS clinical variables. The original manuscript did not state DMT use explicitly and that has been corrected.

Location: Study arms 2 to 3, pages 16 to 17.

Comment 3

Reviewer comment: Provide greater detail on how feasibility of FDSV screening in neurology research will be evaluated within the data analysis plan, including safe disclosure, safety management, and support pathways.

Response: The quantitative analysis plan has been expanded to define feasibility using descriptive process and safety indicators. Indicators cover: safe approach and uptake; safe disclosure setting; screening paused or discontinued due to distress or safety concerns; and the frequency and type of safety management and support pathway actions initiated following screening. Denominators are specified for each indicator.

Location: Feasibility, safe disclosure, and response pathways, pages 17 to 18.

Comment 4

Reviewer comment: Elaborate on the calculation of the sample size in the quantitative arm of the research.

Response: Quantitative sample size is now justified as feasibility based. The screening target of approximately 1,000 is derived from annual clinic throughput of about 1,000 individual people with MS over a 12 month recruitment window, noting some attend more than once, and constrained by the requirement to approach only when the person is private and alone. The survey target of approximately 100 is stated as pragmatic expert consensus on likely uptake for a sensitive topic, and the protocol specifies an anticipated range of 70 to 150 completed surveys. To contextualise feasibility precision, the protocol states that with n≈1,000 screened the 95% CI half width for a binary feasibility measure is typically up to ±3.1%, and with n≈100 survey respondents up to ±9.8% (worst case when the observed value is near 50%).

Location: Sample size, pages 11 to 12.

Comment 5

Reviewer comment: Describe how thematic saturation will be assessed in the qualitative arm, including how thematic saturation is defined and calculated. Clarify the rationale for the target number of approximately 100 surveys and specify any estimated range for survey numbers.

Response: The protocol defines thematic saturation as the point at which additional interviews do not generate new substantive codes or concepts that modify the coding framework or alter interpretation of themes relevant to the aims, including domains aligned with MARAM, LIVES, and condition specific features. Saturation will be assessed iteratively within each participant group during data collection and analysis. An audit trail will be maintained, including dated codebook versions, analytic memos, and analysis meeting notes documenting when codes are added, merged, or refined and the rationale for changes. A minimum of approximately 30 interviews is specified to support depth and diversity across participant groups and key domains. Additional interviews will be undertaken only if the audit trail indicates continued emergence of new substantive concepts or insufficient representation across key domains required to address the aims. The survey target and range have been clarified as described under Comment 4.

Location: Qualitative sample size and analytic sufficiency, pages 11 to 12. Survey target and range, pages 11 to 12.

Comment 6

Reviewer comment: Outline the approach to managing missing or incomplete data, given the potential for substantial missingness in an extensive protocol.

Response: A missing data statement has been added to the quantitative analysis plan. Missingness will be reported by study component and variable, with denominators provided for all estimates. Missingness for routinely captured MS variables is anticipated to be low based on established site recording practices, whereas survey item non response is uncertain given the sensitive content and the option to skip questions, and will be interpreted as a feasibility and acceptability finding. No imputation will be undertaken. Sensitivity analyses will be conducted where appropriate.

Location: Quantitative data analysis, pages 18 to 19.

Comment 7

Reviewer comment: Expand the Background and Context section to include results from other studies worldwide on IPV and FDSV among people with MS to strengthen the rationale.

Response: The Background and Research Context sections have been expanded to summarise the international MS literature on IPV and FDSV, including study settings, measurement approaches, and prevalence estimates. The revision includes recent multicentre data from Italy (2026), alongside existing evidence from the United States, Iran, Sweden, and Norway.

Location: Research Context, pages 4 to 6.

---

## [Decision Letter · Decision Letter 1]

25 Feb 2026

Study protocol for a mixed methods convergent investigation of family domestic and sexual violence in multiple sclerosis and broader neurology in Australia

Version 1 (2025) dated 01 July 2025.

PONE-D-25-65254R1

Dear Dr. Nesbitt,

We’re pleased to inform you that your manuscript has been judged scientifically suitable for publication and will be formally accepted for publication once it meets all outstanding technical requirements.

Kind regards,

Yordanis Enríquez Canto, Ph.D.

Academic Editor

PLOS One

Additional Editor Comments (optional):

Thank you for your thorough revisions and for addressing the reviewers’ feedback. I am pleased to recommend your manuscript for acceptance. However, I noticed that the current title includes “Version 1 (2025) dated 01 July 2025.” This section should be omitted in the final published title, which should end at “…in Australia.”

Please update the title accordingly before publication.

Reviewers' comments:

Reviewer's Responses to Questions

**Comments to the Author**

1. Does the manuscript provide a valid rationale for the proposed study, with clearly identified and justified research questions?

Reviewer #1: Yes

Reviewer #2: Yes

2. Is the protocol technically sound and planned in a manner that will lead to a meaningful outcome and allow testing the stated hypotheses?

Reviewer #1: Yes

Reviewer #2: Yes

3. Is the methodology feasible and described in sufficient detail to allow the work to be replicable?

Reviewer #1: Yes

Reviewer #2: Yes

4. Have the authors described where all data underlying the findings will be made available when the study is complete?

Reviewer #1: Yes

Reviewer #2: Yes

5. Is the manuscript presented in an intelligible fashion and written in standard English?

Reviewer #1: Yes

Reviewer #2: Yes

You may also provide optional suggestions and comments to authors that they might find helpful in planning their study.

Reviewer #1: All questions/comments have been appropriately addressed by the authors. I look forward to reading about the results from the eventual study.

Reviewer #2: The study is well-positioned to generate important evidence in such under-researched area, and the protocol, as revised, is now suitable for publication. No further reviewer input is required from my perspective.

**Do you want your identity to be public for this peer review?** For information about this choice, including consent withdrawal, please see our Privacy Policy

Reviewer #1: No

Reviewer #2: No

---

## [Editor Report · Acceptance letter]

PONE-D-25-65254R1

PLOS One

Dear Dr. Nesbitt,

I'm pleased to inform you that your manuscript has been deemed suitable for publication in PLOS One. Congratulations! Your manuscript is now being handed over to our production team.

Kind regards,

on behalf of

Prof. Yordanis Enríquez Canto

Academic Editor

PLOS One